

# Corallimorpharians are not "naked corals": insights into relationships between Scleractinia and Corallimorpharia from phylogenomic analyses

Mei Fang Lin[1,2], Wen Hwa Chou[3], Marcelo V. Kitahara[4,5], Chao Lun Allen Chen[3], David John Miller[1,2] and Sylvain Forêt[1,6]

[1] Australian Research Council Centre of Excellence for Coral Reef Studies, James Cook University, Townsville, QLD, Australia
[2] Comparative Genomics Centre and Department of Molecular and Cell Biology, James Cook University, Townsville, QLD, Australia
[3] Biodiversity Research Center, Academia Sinica, Taipei, Taiwan
[4] Departamento de Ciências do Mar, Universidade Federal de São Paulo, Santos, São Paulo, Brazil
[5] Centro de Biologia Marinha, Universidade Federal de São Paulo, São Sebastião, São Paulo, Brazil
[6] Research School of Biology, Australian National University, Canberra, ACT, Australia

Corresponding authors
David John Miller,
david.miller@jcu.edu.au
Sylvain Forêt,
sylvain.foret@anu.edu.au

## ABSTRACT

Calcification is one of the most distinctive traits of scleractinian corals. Their hard skeletons form the substratum of reef ecosystems and confer on corals their remarkable diversity of shapes. Corallimorpharians are non-calcifying, close relatives of scleractinian corals, and the evolutionary relationship between these two groups is key to understanding the evolution of calcification in the coral lineage. One pivotal question is whether scleractinians are a monophyletic group, paraphyly being an alternative possibility if corallimorpharians are corals that have lost their ability to calcify, as is implied by the "naked-coral" hypothesis. Despite major efforts, relationships between scleractinians and corallimorpharians remain equivocal and controversial. Although the complete mitochondrial genomes of a range of scleractinians and corallimorpharians have been obtained, heterogeneity in composition and evolutionary rates means that mitochondrial sequences are insufficient to understand the relationship between these two groups. To overcome these limitations, transcriptome data were generated for three representative corallimorpharians. These were used in combination with sequences available for a representative range of scleractinians to identify 291 orthologous single copy protein-coding nuclear markers. Unlike the mitochondrial sequences, these nuclear markers do not display any distinct compositional bias in their nucleotide or amino-acid sequences. A range of phylogenomic approaches congruently reveal a topology consistent with scleractinian monophyly and corallimorpharians as the sister clade of scleractinians.

## INTRODUCTION

Scleractinian corals are the subject of intense scientific, public and, therefore, media interest, particularly because of the uncertain fate of coral reefs in the face of ever increasing anthropogenic challenges (*Done, 1999*; *Hughes, 2003*; *Hughes et al., 2003*). Due to their capacity to deposit massive continuous calcareous skeletons, the coral reef framework built by scleractinians provides one of the most complex and diverse of biological habitats (*Cohen & Holcomb, 2009*). Despite their ecological importance and our economic dependence on them (*Moberg & Folke, 1999*), we know remarkably little about the evolutionary history of this animal group. This lack of understanding limits our ability to predict how corals, and therefore the diverse habitats that they support, will respond to climate change and ocean acidification (OA).

Although the vast majority of scleractinian fossils post-date the sudden appearance of diverse coral families 14 My after the Permian/Triassic boundary, there is now evidence that the evolutionary origin of the group is rooted deep in the Paleozoic. In brief, molecular clock estimates calibrated using the earliest fossils that can be unambiguously assigned to extant clades, and whose unique skeletal characters can be unequivocally recognized in fossil coralla, imply that the scleractinian corals originated from a non-skeletonized ancestor in the Ordovician (*Stolarski et al., 2011*). When considered in conjunction with the elusive Paleozoic fossil record of the scleractinian lineage (*Ezaki, 1997*; *Ezaki, 2000*; *Scrutton & Clarkson, 1991*), this suggests that either the fossil record for the period between the Ordovician and late Permian is yet to be discovered, or that skeleton formation may be an ephemeral trait within the Scleractinia (*Stanley & Fautin, 2001*).

The idea that the ability of corals to deposit a skeleton may be an ephemeral trait on evolutionary time scales, the presence or absence of a calcareous skeleton potentially reflecting prevailing environmental conditions, together with the anatomical similarity of Actiniaria, Corallimorpharia, and Scleractinia (*Daly et al., 2007*; *Stanley & Fautin, 2001*), led Stanley (*Stanley, 2003*) to propose the "naked coral" hypothesis (Fig. 1A). The central idea of this hypothesis is that "different groups of soft-bodied, unrelated anemone-like anthozoans gave rise to various calcified scleractinian-like corals through aragonitic biomineralization" (*Stanley, 2003*), potentially explaining the sudden appearance of a diverse and differentiated range of scleractinian skeletal types in the Triassic. Under this hypothesis, the scleractinian skeleton is not a synapomorphy, but stands for an organization grade. Consistent with this hypothesis, the Scleractinia were paraphyletic in molecular phylogenetic analyses based on amino acid (aa) sequence data from mitochondrial protein-coding genes (*Medina et al., 2006*). In these analyses, it was estimated that corallimorpharians—anthozoans without a skeleton—diverged from the Robust scleractinian clade during the late- and mid-Cretaceous, implying that corallimorphs were descended from a coral that had undergone skeleton loss during a period of increased ocean acidification. Whilst ocean acidification events occurred in that period they did not cause any reef crisis (*Honisch et al., 2012*; *Kiessling, Simpson & Foote, 2010*; *Pandolfi et al., 2011*). Moreover, some alternative phylogenetic analyses based on a range of other molecular markers (*Chen, Wallace & Jackie, 2002*; *Fukami et al., 2008*; *Lin*

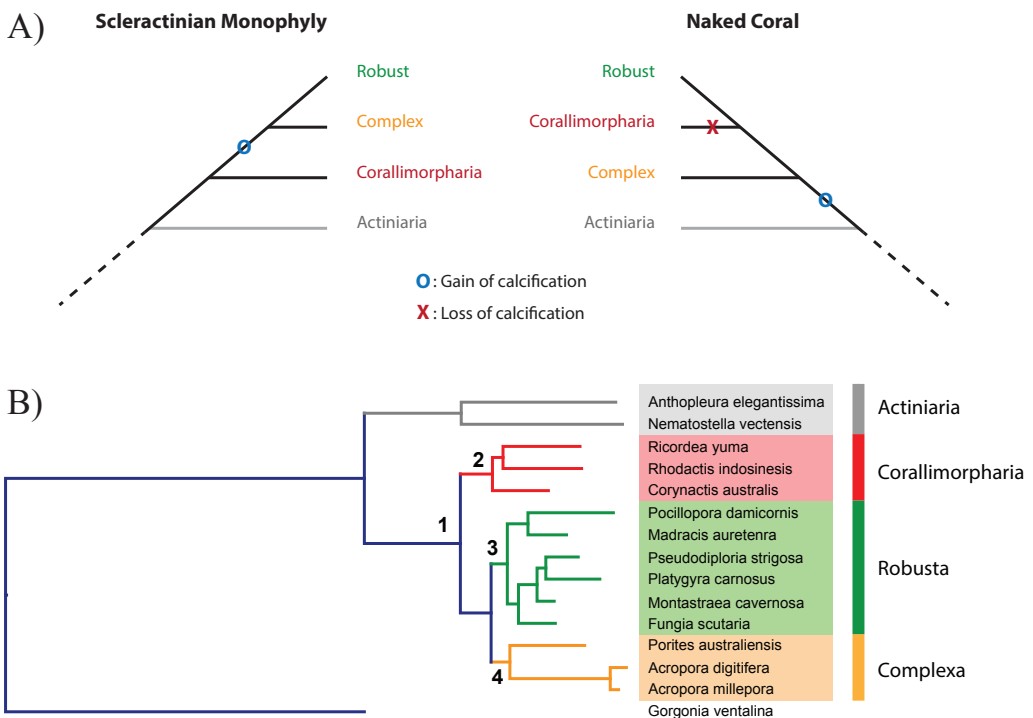

**Figure 1** **Relationship between Scleractinia and Corallimorpharia.** (A) The two competing hypotheses regarding the relationship between Corallimorpharia and Scleractinia: scleractinian monophyly and the "naked coral" topology. Scleractinian monophyly implies that the ability to calcify was acquired in the ancestor of Scleractinia, whilst the naked coral hypothesis requires secondary loss of this trait in the ancestor of Corallimorpharia. (B) Maximum likelihood phylogenetic tree based on the amino acid sequences of 291 nuclear genes from 15 anthozoans with the JTT + GAMMA + I model. The critical nodes (1, 2, 3 and 4) are fully supported, as reported in Table 1. The same topology was obtained for all the other analyses with equally strong support (see Table 1).

*et al., 2014*; *Romano & Palumbi, 1996*) did not support the naked coral senario, and it has become apparent that mitochondrial sequence data might not be appropriate for the elucidation of phylogenetic relationships within the Corallimorpharia/Scleractinia clade (*Kitahara et al., 2014*).

The issue of coral/corallimorpharian relationships is of particular importance, because the idea that skeleton loss can occur as a consequence of ocean acidification carries implications for the future of corals and coral reefs under climate change and elevated atmospheric [$CO_2$]. A better understanding of coral evolution more broadly has further implications for coral reef futures. For instance, it is important to understand how prior OA events (*Kiessling, Simpson & Foote, 2010*) have impacted the scleractinian lineage, and the underlying causes of previous "reef crises" (*Hoegh-Guldberg et al., 2007*).

In order to shed light on the relationship between Corallimorpharia and Scleractinia, phylogenomic analyses were carried out based on 291 single-copy nuclear protein-coding genes from a representative range of robust and complex corals, corallimorpharians, and sea anemones. To enable the phylogenomic analysis, it was first necessary to generate

comprehensive transcriptome assemblies for the three corallimorpharian species—
*Rhodactis indosinensis, Ricordea yuma* and *Corynactis australis*. These molecular data
constitute an important resource for this neglected animal group. Analyses were carried out
both at the amino acid and nucleotide levels on concatenated, partitioned and individual
alignments using multiple inference methods. The results provide strong support for
scleractinian monophyly, allowing rejection of the hypothesis that corallimorpharians are
"naked corals"—descendants of a scleractinian that had undergone skeleton loss.

## MATERIALS AND METHODS

### Transcriptome assembly and data matrix

The complete workflow from data collection to analysis is summarized in Fig. S1. The
taxonomic sampling (Table S1) included three "Complex" corals from two families,
six "Robust" corals from five families, three corallimorpharians representing three
families, and two actiniarians. *Gorgonia ventalina* was used as the outgroup. The
new corallimorpharian transcriptome data were obtained from two zooxanthellate
species (*Ricordea yuma* and *Rhodactis indosinensis*) and an azooxanthellate species
(*Corynactis australis*). *Ricordea yuma* samples were collected from the Great Barrier
Reef (18°25′35.20″S, 146°41′10.91″E). *Corynactis australis* colonies were collected from
Jervis Bay, New South Wales (35°4′14.11″S, 150°41′48.20″E). The *Rhodactis indosinensis*
samples were collected at Beitou fishing harbor, Keelung, Taiwan. The transcriptomes
were generated from purified RNA extracted by using Trizol Reagent (Invitrogen, USA)
and dissolved in RNase-free water. High throughput sequencing was conducted using
the Illumina HiSeq 2000 platform. The transcriptomes were then assembled with Trinity
(r2013_08_14) using default settings (*Grabherr et al., 2011*). *Symbiodinium* sequences
were eliminated using PSyTranS (https://github.com/sylvainforet/psytrans). The resulting
contigs were clustered with CD-HIT-EST (*Li & Godzik, 2006*) at a sequence similarity
threshold of 0.9. The contigs were then translated into amino acid sequences with
TransDecoder (*Grabherr et al., 2011*). A summary of the resulting transcritpome assemblies
is given in Table S2. The raw reads and the transcriptome assemblies have been deposited
to NCBI under BioProject PRJNA313487.

HaMStR v13.2 (*Ebersberger, Strauss & Von Haeseler, 2009*) was used to search
for orthologs using three available cnidarian genomes as primer taxa, *Acropora
digitifera* (*Shinzato et al., 2011*), *Nematostella vectensis* (*Putnam et al., 2007*) and *Hydra
magnipapillata* (*Chapman et al., 2010*), with *A. digitifera* as the reference taxon, resulting
in 1,808 core orthologs. The same program was used for an extended search for orthologs
in the other scleractinians, corallimorpharians, and actiniarian transcriptomes. The
*H. magnipapillata* sequences were excluded from the phylogenetic analyses due to their very
high divergence with the Anthozoan sequences. In the end, we identified 291 one-to-one
orthologs across all 15 taxa. Each orthologous group was annotated according to the best
blast hit of the *A. digitifera* protein in that cluster against the NCBI nr database with an
e-value cut-off of 1e-5.

The amino acid (aa) sequences from the 291 orthologous genes were aligned using
MAFFT L-INSI v7.13 (*Katoh & Standley, 2013*) and subsequently trimmed using trimAl

v1.2 with the Heuristic method (*Capella-Gutierrez, Silla-Martinez & Gabaldon, 2009*). The nucleotide (nt) alignments were deduced from the aa alignments as described in (*Kitahara et al., 2014*). The saturation at each nucleotide position was estimated with DAMBE v5.3.110 (*Xia, 2013*), revealing no significant saturation in the dataset (Table S3).

### Supermatrix phylogeny

For the concatenated aa matrix, the best fitting model determined using ProtTest v3 (Darriba et al. 2011) was JTT + G + I. Maximum likelihood (ML) analyses were carried out with RAxML v7.2.6 (*Stamatakis, 2006*) using rapid bootstrapping (-f a). Phylogenies based on the supermatrix were also computed using Bayesian inference (BI) with PhyloBayes MPI v1.5a (*Lartillot et al., 2013*) using the JTT + G + I model. Identical topologies were recovered with CAT-Possion, and CAT + GTR models. Each run contained four chains and ran until convergence. Convergence was assessed after a burn-in period of 2,000 generations following the author's guidelines (maxdiff > 0.1 and effective size > 300). The best fitting model for the nt alignment determined by jModelTest 2 (*Darriba et al., 2012*) was the GTR + G + I, and the phylogenetic inference was carried out in a similar way as the aa analysis. Trees and alignments have been deposited to TreeBase (ID 19254).

### Partitioned phylogeny

Partitions and their corresponding best-fitting models were identified using PartitionFinder (*Lanfear et al., 2012*) with the relaxed clustering algorithm, checking the top 1% schemes. ML analysis was conducted on the partitioned datasets using RAxML v7.2.6 (*Stamatakis, 2006*) with 100 bootstrap replicates. The partitions identified by Partition finder were also used for Bayesian inference using MrBayes v3.2.3 (*Ronquist & Huelsenbeck, 2003*) with 4 runs, 2 million generations saving topologies each 1,000 generations and discarding the first 25% generations as burn-in.

### Concordance factor estimation

Concordance factors were estimated on the 291 individual topologies inferred by MrBayes v3.2.3 (*Ronquist & Huelsenbeck, 2003*) (four runs, 2 million generations 25% burn-in) using BUCKy (*Ane et al., 2006*) with default settings ($\alpha = 1$).

## RESULTS AND DISCUSSION

### Analyses of the concatenated nucleotide supermatrix

The data matrix for the 15 taxa (Table S1) comprised 291 nuclear protein-coding genes, 263 of which have functional annotations, the other 28 coding for unknown proteins (Table S5) that probably correspond to cnidarian- or anthozoan-specific genes. The final alignment of the nucleotide sequences contained 370,809 positions, around 30 times longer than in the previously published phylogenies based on whole mitochondrial genomes (*Kitahara et al., 2014*). A ML phylogeny was inferred using the best fitting model (GTR + G + I), resulting in a topology consistent with scleractinian monophyly and with a bootstrap support of 100% for every node (Table 1). This result is consistent with analyses based on the nucleotide sequences of mitochondrial protein-coding genes (*Fukami et al., 2008*; *Kitahara et al., 2010*; *Kitahara et al., 2014*).

Lin et al. (2016), *PeerJ*, DOI 10.7717/peerj.2463

**Table 1** **Support values of critical nodes (see Fig. 1B for the numbering of nodes) for ML and BI analyses inferred using unpartitioned and partitioned phylogenetic analyses of amino acid and nucleotide data.** The best fitting substitution model for each concatenated unpartitioned dataset is indicated; the best fitting models for the partitioned phylogenies are detailed in Table S4. Scleractinian monophyly is fully supported by all the analyses.

| Data type | Amino acid dataset | | | | Nucleotide dataset | | | | |
| --- | --- | --- | --- | --- | --- | --- | --- | --- | --- |
| Method | Maximum likelihood analyses | | Bayesian inference | | Maximum likelihood analysis | | | Bayesian inference | |
| Supporting value | Bootstrap support(%) | | Posterior probability | | Bootstrap support(%) | | | Posterior probability | |
| Matrix type | Concatenated matrix | Partitions | Concatenated matrix | Partitions | Concatenated matrix | Partitions by gene | Partition by codon | Partitions by gene | Partition by codon |
| Selected nodes/ substition model | JTT + G + I | 153 subset | JTT + G + I | 15 | GTR + G + I | 75 subsets | 106 subsets | 75 subsets | 106 subsets |
| 1 (Corallimorpharia, Scleractinia) | 100 | 100 | 1 | 1 | 100 | 100 | 100 | 1 | 1 |
| 2 (Corallimorpharia) | 100 | 100 | 1 | 1 | 100 | 100 | 100 | 1 | 1 |
| 3 (Robusta | 100 | 100 | 1 | 1 | 100 | 100 | 100 | 1 | 1 |
| 4 (Complexa) | 100 | 100 | 1 | 1 | 100 | 100 | 100 | 1 | 1 |
To take into account the fact that different regions of the alignment can evolve at different rates and according to different models, the aa supermatrix was partitioned using PartitionFinder, by gene and by codon, resulting in 75 and 106 partitions respectively. ML and BI phylogenies were then inferred for each partitioning scheme, all strongly supporting scleractinian monophyly (Table 1). Thus, both unpartitioned and partitioned analyses of the nucleotide supermatrix consistently support the monophyly of Scleractinia. These findings corroborate a number of previous studies (e.g., *Fukami et al., 2008*; *Kitahara et al., 2010*; *Kitahara et al., 2014*; *Lin et al., 2014*; *Stolarski et al., 2011*). However, as analyses of mitochondrial protein-coding sequences at the amino acid and nucleotide levels result in distinct tree topologies (*Kitahara et al., 2014*; *Medina et al., 2006*), ML and BI analyses were also conducted based on the aa sequences of the nuclear protein-coding genes.

## Analyses of the concatenated amino acid supermatrix

The concatenated amino-acid alignment consisted of 122,170 positions. Both ML and BI methods generated phylogenetic trees in which all nodes were strongly supported (Table 1). In the ML reconstruction, all the bootstrap values were >70% and most nodes had 100% support. In the BI analysis, the posterior probability for all the nodes was 100%. Partitioning of the amino acid alignment resulted in 153 subsets. ML and BI phylogenies were then inferred based on the best substitution model for each partition (Table S6) and also strongly supported the monophyly of scleractinians (Table 1). In summary, unpartitioned and partitioned analysis of nuclear markers at the amino-acid and nucleotide level are congruent. The major implication of these analyses of nuclear sequence data is that corallimorpharians are not scleractinians that have undergone skeleton loss (Fig. 1A). However, a ML tree based on the mitochondrial proteins of a set of species close to those used for nuclear markers recovered the naked coral topology (Fig. S2), consistent with the results reported by *Kitahara et al. (2014)*, which could be a result of the sequence composition biases in these mitochondrial genomes.

## Sequence composition

In the case of mt genomes, significant differences in the base composition of protein coding genes were observed between corallimorpharians, robust and complex corals, resulting in different patterns of codon usage and amino acid composition across the various lineages (*Kitahara et al., 2014*). In order to investigate the potential for compositional bias to affect the topology recovered for nuclear protein-coding genes, base composition was estimated for each of the 15 taxa included in the present analyses (Table S4). Base composition was generally similar across all the hexacorallian groups, but the octocoral (A + T) content (57.96%) was significantly higher. Within the Hexacorallia, the complex scleractinian clade had the highest (A + T) content (56.5%) and, consequently, a higher proportion of (A + T)-rich aa (FYMINK). The remaining groups (i.e., Actiniaria and Scleractinia [Robusta clade]) displayed an overall (A + T) content between 55.00 and 55.95% and no major differences between FYMINK and (G + C)-rich aa (GARP) (Fig. S3). The thymine and cytosine contents of nuclear protein coding genes of Robusta differed slightly (<1%) across all three codon positions compared to other scleractinians,. The nuclear

protein-coding genes of Actinaria, Corallimorpharia and Scleractinia have therefore a very similar composition (Fig. S3). In comparison to proteins encoded by the mitochondrial genome, nuclear-encoded proteins of anthozoans contain, in general, more lysine (7% vs 2%), aspartic acid (5.5% vs 2%), and glutamic acid (7% vs 2.5%) residues, but significantly less phenylalanine (4% vs 8%, and 13% in robust corals).

Major differences in the composition of mitochondrial protein-coding genes at both the nucleotide and amino acid levels support the idea that the mitochondrial genomes of robust corals are evolving at a different rate to those of other hexacorallians (*Aranda et al., 2012*; *Fukami & Knowlton, 2005*; *Kitahara et al., 2014*). However, no such compositional biases appear to hold for nuclear protein-coding genes, implying that these nuclear sequences are more appropriate sources of phylogenetic information than mitochondrial data (*Kitahara et al., 2014*).

## Analysis of individual gene topologies

Genes at different genomic locations may have distinct evolutionary histories, and thus different topologies may be recovered (*Akanni et al., 2014*; *Ane et al., 2006*; *Pisani, Cotton & Macinerney, 2007*). We constructed trees based on individual genes using ML and BI and explored the distribution of the various topologies. For all types of inference, scleractinian monophyly was recovered by the majority of genes, while only a small proportion of the trees were concordant with the naked coral hypothesis (Fig. 2A). The patterns of topologies for each tree across all types of inference was then investigated. Again, the most common pattern was genes producing scleractinian monophyly across all types of inference, while only a few genes were consistent with the naked coral hypothesis for all the reconstruction methods (Fig. 2B). A sizeable proportion of genes did not agree with either scenario, as can be expected when inferring such deep relationships based on single markers. A search for systematic differences between the genes supporting the two competing of topology did not reveal any differential Gene Ontology enrichment. However, genes supporting the naked coral topology for all types of inference were found to be significantly shorter than genes supporting the alternative topology (Fig. 2C). This suggests that genes supporting the naked coral hypothesis might be too small for the inference of the correct topology.

Bayesian Concordance Analysis was then used to evaluate the contribution of individual genes to the final topology in the BI trees. High concordance factor (CF) values on branches indicate support from multiple genes (*Weisrock, 2012*). The primary topology recovered by concordance analysis supported scleractinian monophyly (Fig. S4). In particular, the branches descending from the split between Scleractinia and Corallimorpharia have mean sample-wise CF values of 0.627 and 0.756 respectively. This result is consistent with the unpartitioned and partitioned analyses of the concatenated sequence data and therefore indicates broad support across the sampled nuclear genes for the monophyly of scleractinians.

## Are corallimorpharians naked corals?

Comparisons based on mitochondrial genomic architecture (*Lin et al., 2014*) suggest that corallimorpharians are derived from an azooxanthellate ancestor. Anatomical similarities

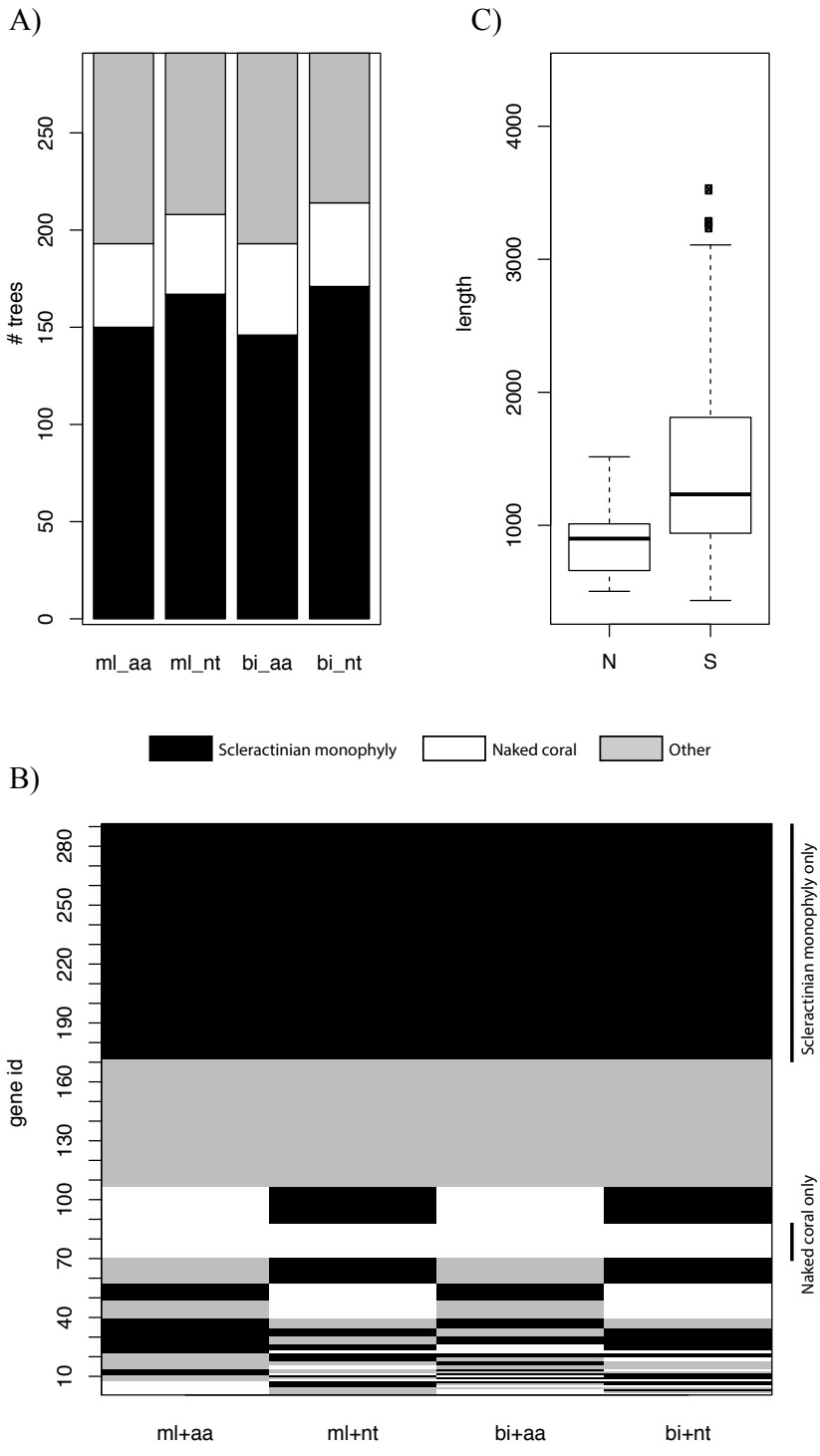

**Figure 2** (A) Numbers of trees based on individual genes supporting scleractinian monophyly, naked coral or other topologies for maximum likelihood (ml) and Bayesian inference (bi) for amino-acid (aa) and nucleotide alignments (nt). The majority of trees are consistent with scleractinian monophyly, whereas few support the naked corals scenario. (B) Summary of the concordance of phylogenetic inference for each gene. Each line represents a gene. (continued on next page...)

**Figure 2 (…continued)**
The main pattern (black lines) represents genes that are fully consistent with scleractinian monophyly, whilst only a small proportion of genes consistently agree with the naked coral hypothesis (white lines). The checkered lines correspond to genes producing topologies that are not consistent across the different types of inference. The patterns are sorted from most abundant at the top of the figure to least abundant at the bottom. (C) Distribution of sequence lengths for genes consistent with scleractinian monophyly (S, black lines in B) and the naked coral hypothesis (N, white lines B). The sequences of the genes consistent with scleractinian monophyly are significantly longer (Mann Whitney $U$ test $p = 0.0004$).

between scleractinians and corallimorpharians support a close relationship between them, but corallimorpharians not only lack mineralized skeletons, but also differ from scleractinians in terms of several characters—for example: the condition of the mesoglea, tentacular arrangement and the presence of homotrichs in the tentacles (*Den Hartog, 1980*). Systematically, the taxonomic rank of Corallimorpharia has been controversial (*Budd et al., 2010*; *Daly, Fautin & Cappola, 2003*; *Den Hartog, 1980*; *Medina et al., 2006*; *Romano & Cairns, 2000*). The phylogenomic analyses presented here provide strong support for scleractinian monophyly, and allow rejection of the idea that the corallimorpharian lineage was derived from corals by skeleton loss. The analyses supporting this latter idea were based on amino acid sequence data from mitochondrial genomes (*Medina et al., 2006*), but it is now clear there are fundamental problems in using mitochondrial data to infer phylogenetic relationships amongst hexacorallians (*Kitahara et al., 2014*), as is also the case in beetles (*Sheffield et al., 2009*) and some groups of mammals (*Huttley, 2009*).

## Insights into coral evolution

Taking into account the fossil and published molecular data (*Kiessling, Simpson & Foote, 2010*; *Stolarski et al., 2011*), the analyses above imply that the ability to secrete a skeleton was acquired early in scleractinian evolution, but was followed by multiple origins of skeleton complexity in various subclades (*Romano & Cairns, 2000*). The Paleozoic fossil record (*Ezaki, 1997*; *Ezaki, 1998*; *Scrutton, 1993*; *Scrutton & Clarkson, 1991*) and molecular data (*Stolarski et al., 2011*) both imply that the earliest scleractinians were solitary and inhabited deep water and therefore lacked photosynthetic symbionts. The sudden appearance of highly diversified forms of Scleractinia about 14 Ma after the end-Permian extinction (the "Great Dying" (*Roniewicz & Morycowa, 1993*; *Stanley, 2003*; *Veron, 1995*; *Wells, 1956*)) might be explained by multiple independent origins from deep-water ancestors (e.g., the family Agariciidae (*Kitahara et al., 2012*)). It thus appears likely that the acquisition of photosynthetic symbionts and the development of coloniality have probably both occurred independently on multiple occasions (but see *Barbeitos, Romano & Lasker, 2010*), resulting not only in a wide range of skeletal phenotypes but also in habitat expansion, which has played important roles in the formation of shallow-water reefs.

It has been demonstrated that, when maintained under acidic conditions (pH7.3–7.6), at least some corals can survive for 12 months after undergoing skeleton loss, recovering fully after return to normal seawater (*Fine & Tchernov, 2007*). One interpretation of these experiments is that, during evolution, the coral lineage might have been able to alternate between soft and skeletonized forms, potentially explaining the gaps in the fossil record. However, the fact that corallimorpharians are not derived from corals, and the monophyly

of extant Scleractinia, suggests otherwise—that skeleton-less corals are not viable on evolutionary time scales. This has important implications for the future of the coral lineage—the evolutionary resilience of the Scleractinia may have depended in the past on deep sea refugia, as most of the ''reef crises'' have coincided with rapid increases in both OA and sea surface temperature (*Pandolfi et al., 2011*). Deep-sea corals would have escaped the challenges of high SST, thus the coral lineage may have been able to re-establish itself in the shallows when more favourable conditions returned. At the present time, unprecedented rates of increase in OA and SST are occurring concurrently with massive disruption of deep-sea habitats caused by deep sea trawling, prospecting and mining (*Guinotte et al., 2006*; *Ramirez-Llodra et al., 2011*; *Barbier et al., 2014*). Is the resilience of the Scleractinia as a lineage therefore at risk?

## ACKNOWLEDGEMENTS

The authors thank Allen Collins and an anonymous reviewer for their valuable comments on the manuscript, Mr. Chris Benstead (Reef HQ Aquarium), Dr. Joshus Madin (Macquarie University), Mr. Chao-Yang Kuo (James Cook University) for the assistance with sampling corallimorpharians and field logistics.

### Funding

This work was supported by ARC CoE CE140100020. MVK received postdoctoral support (FAPESP # 2011/17537-1) from the São Paulo Research Foundation. The funders had no role in study design, data collection and analysis, decision to publish, or preparation of the manuscript.

### Grant Disclosures

The following grant information was disclosed by the authors:
ARC CoE: CE140100020.
São Paulo Research Foundation: FAPESP # 2011/17537-1.

### Competing Interests

The authors declare there are no competing interests.

### Author Contributions

- Mei Fang Lin and Sylvain Forêt conceived and designed the experiments, performed the experiments, analyzed the data, contributed reagents/materials/analysis tools, wrote the paper, prepared figures and/or tables, reviewed drafts of the paper.
- Wen Hwa Chou performed the experiments, analyzed the data, contributed reagents/materials/analysis tools, reviewed drafts of the paper.
- Marcelo V. Kitahara and David John Miller conceived and designed the experiments, analyzed the data, contributed reagents/materials/analysis tools, wrote the paper, reviewed drafts of the paper.

- Chao Lun Allen Chen conceived and designed the experiments, performed the experiments, contributed reagents/materials/analysis tools, reviewed drafts of the paper.

## DNA Deposition

The following information was supplied regarding the deposition of DNA sequences:

The raw reads and the transcriptome assemblies have been deposited to NCBI under BioProject PRJNA313487.

## Data Availability

TreeBASE: ID 19254.

## Supplemental Information

Supplemental information for this article can be found online at http://dx.doi.org/10.7717/peerj.2463#supplemental-information.

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
