# Peer review of "Corallimorpharians are not “naked corals”: insights into relationships between Scleractinia and Corallimorpharia from phylogenomic analyses"

_PeerJ, doi:10.7717/peerj.2463_

## Round 0.1 · original submission · Minor Revisions

The reviewers concurred that this study is a great contribution to the field and deserves publication. There are just a few minor suggestions recommended by one of the reviewers and I would encourage the authors to address them. A revised version is needed before we can accept the paper for publication.

Reviewer 1 ·

Basic reporting

No Comments

Experimental design

Here I feel I must admit to having no experience in analysis of transcriptome data, thus am not in the position to detect a flaw in the analysis. However, the authors have in my opinion done a thorough job explaining the methodology and the data appear sound.

Validity of the findings

No Comments

Additional comments

It is so refreshing to receive a manuscript for review that is so well conceived and written that I have no substantive criticisms to offer, but only a spelling error and a couple of misplaced words! I look forward to discussing this work with students and colleagues when it is published.

Annotated reviews are not available for download in order to protect the identity of reviewers who chose to remain anonymous.

·

Basic reporting

No Comments

Experimental design

Excellent.

Validity of the findings

No Comments

Additional comments

This is an excellent piece of work that should be published in PeerJ after very minor revisions. The study convincingly shows that genomic data strongly support the hypothesis that Scleractinia is monophyletic and does not include Corallimorpharia. I think the authors somewhat overplay the idea that present knowledge on the subject is equivocal given their excellent work already published falsifying the naked coral hypothesis. That said, this is the first evidence from phylogenomic analyses and it is welcomed and convincing.

Introduction
“Due to their unique capacity to deposit massive continuous calcareous skeletons”; this capacity is not unique to scleractinians.

“we know remarkably little about the evolutionary history of this animal group.” Do the authors really believe this, after they have made so many contributions toward understanding this very subject?

“had undergone skeleton loss during a period increased ocean acidification.” ==> “during a period of increased ocean acidification.”

“events occurred in that period but they did not cause any” ==> “events occurred in that period, they did not cause any”

“understand, how prior OA events” ==> “understand how prior OA events”

Materials and Methods:
“TheRhodactis indosinensis samples” add space and italics on species name.

Insights into coral evolution:
“the acquisition of photosynthetic symbionts and the development of coloniality have both occurred independently on multiple occasions,” Barbeitos et al. 2010 came to a different conclusion based on ancestral state reconstructions, e.g., “Using Bayesian ancestral state reconstruction, we found that symbiosis was lost at least three times and coloniality lost at least six times, and at least two instances in which both characters were lost.” Is it possible that coloniality evolved, then symbiosis, and then later still independent losses of symbiosis and coloniality? Perhaps the taxon sampling in this study is insufficient to really test these hypotheses, and I would suggest that the Barbeitos et al. study be cited as an alternative view on this topic.

“However, the fact that corallimorpharians are not derived from corals, and the monophyly of extant Scleractinia, implies otherwise - that skeleton-less corals are not viable on evolutionary time scales.” I am not sure this follows. It would just mean that there is no evidence that skeleton-less corals are viable on evolutionary time scales.” This is a classic absence of evidence argument; this is not evidence for absence of the stated viability.

Conclusion:
“Scleractinia may have depended in the past on deep sea refugia, as most of the “reef crises” have coincided with rapid increases in both OA and sea surface temperature (Pandolfi et al. 2011). Deep-sea corals would have escaped the challenges of high SST, thus the coral lineage may have been able to re-establish itself in the shallows when more favourable conditions returned. At the present time, unprecedented rates of increase in OA and SST are occurring concurrently with massive disruption of deep-sea habitats (Guinotte et al. 2006; Ramirez-Llodra et al. 2011). Is the resilience of the Scleractinia as a lineage therefore at risk?”
This mostly makes sense, but the first sentence implicates BOTH OA and SST in reef crises and then later suggests that deep-sea habitats are being disrupted. Many of the disruptions stem from increasing CO2 as noted in the Ramirez-Llodra et al. paper, “increases in atmospheric CO2 and facets and consequences of climate change will have the most impact on deep-sea habitats and their fauna”. It is unclear why this would not have been the case during past reef crises as well. All in all, the final question seems fair. But the complexities of the situation appear to be glossed over in the brief conclusion.

---

## Round 0.2 · accepted · Accept

The authors have addressed satisfactorily the comments and suggestion raised by the reviewers.